# Compensating Circuit to Reduce the Impact of Wire Resistance in a Memristor Crossbar-Based Perceptron Neural Network

**DOI:** 10.3390/mi10100671

**Published:** 2019-10-02

**Authors:** Son Ngoc Truong

**Affiliations:** Faculty of Electrical and Electronics Engineering, Ho Chi Minh City University of Technology and Education, Ho Chi Minh City 70000, Vietnam; sontn@hcmute.edu.vn; Tel.: +84-931-085-929

**Keywords:** memristor, crossbar array, wire resistance, synaptic weight, character recognition

## Abstract

Wire resistance in metal wire is one of the factors that degrade the performance of memristor crossbar circuits. In this paper, an analysis of the impact of wire resistance in a memristor crossbar is performed and a compensating circuit is proposed to reduce the impact of wire resistance in a memristor crossbar-based perceptron neural network. The goal of the analysis is to figure out how wire resistance influences the output voltage of a memristor crossbar. It emerges that the wire resistance on horizontal lines causes the neuron’s output voltage to vary more than the wire resistance on vertical lines. More interesting, the voltage variation caused by wire resistance on horizontal lines increases proportionally to the length of metal wire. The first column has small voltage variation whereas the last column has large voltage variation. In addition, two adjacent columns have almost the same amount of voltage variation. Under these observations, a memristor crossbar-based perceptron neural network with compensating circuit is proposed. The neuron’s outputs of two columns are put into a subtractor circuit to eliminate the voltage variation caused by the wire resistance. The proposed memristor crossbar-based perceptron neural network is trained to recognize the 26 characters. The proposed memristor crossbar shows better recognition rate compared to the previous work when wire resistance is taken into account. The proposed memristor crossbar circuit can maintain the recognition rate as high as 100% when wire resistance is as high as 2.5 Ω. By contrast, the recognition rate of the memristor crossbar without the compensating circuit decreases by 1%, 5%, and 19% when wire resistance is set to be 1.5, 2.0, and 2.5 Ω, respectively.

## 1. Introduction

Neuromorphic computing, inspired from biological perception, was introduced by C. Mead in the late 1980s [1]. It has been expected to become an alternative architecture to overcome the bottleneck of von Neumann computer architectures [1,2]. Neuromorphic computing refers to a hardware implementation of a brain-inspired system, which has the capabilities of parallel processing like a human brain. For realizing neuromorphic computing systems, various research activities, based on CPUs (Central Processing Units), GPUs (Graphics Processing Units), FPGAs (Field-Programmable Gate Arrays), analog circuits, memory circuits, etc., have been proposed in the past two decades [3,4,5,6,7,8]. These architectures are based on CMOS (Complementary-Metal-Oxide-Semiconductor) technology, which is approaching the end of their capabilities because scaling CMOS down faces several fundamental limiting factors stemming from electron thermal energy and quantum-mechanical tunneling [9,10]. The memristor crossbar array has been one of the promising candidates for realizing neuromorphic computing systems because crossbar architecture can be made with high density and low cost [11]. Memristor was postulated by Leon O. Chua in 1971 as the fourth basic circuit element and experimentally demonstrated by HP Lab in 2008 [12,13]. A memristor is a resistor with modifiable resistance, which makes it ideal for mimicking the synaptic plasticity of biological neurons [14]. The early memristor-based synaptic circuits are composed of memristors and CMOS transistors [15,16,17]. However, pure memristor crossbar arrays without CMOS devices seem to be more efficient in terms of their integration and power consumption [18,19,20,21,22,23]. Miao Hu et al. proposed a crossbar synaptic array that is composed of a plus and minus crossbar array representing plus- and minus-polarity connection matrices for analog neuromorphic computing [20]. Such a pure memristor crossbar array is very effective in realizing the bio-inspired systems in term of power consumption and area occupation. To reduce area and power consumption, S. N. Truong proposed a new memristor crossbar array architecture which is composed of a single memristor array and a constant-term circuit [21]. 

In a memristor array, some amount of voltage drop can be caused by interconnect resistance, also known as wire resistance along the row and the column lines [19,24,25,26,27]. Wire resistance degrades the performance of the circuit more seriously when the array size increases [25]. To mitigate the impact of wire resistance, several interesting schemes were proposed [24,25,26,27]. These schemes are effective when they are applied to a memristor crossbar array, in which memristors are used as binary switches between two distinct high and low resistance states (HRS and LRS, respectively). However, the impact of wire resistance in an analog memristor crossbar array for realizing the synaptic weight matrix was not fully considered. In this work, we propose a memristor crossbar array with a compensating circuit for implementing the analog synaptic array of a perceptron neural network. The impact of wire resistance is mitigated by compensating the voltage variation of two adjacent columns. 

In this work, the output voltages of columns are figured out with taking the existing of wire resistance into account. The mathematical analysis and the simulation result show that the output voltage of columns increase, which is caused by the amount of voltage lost from wire resistance. The column close to the first one has a small variation of voltage, compared to the one far from the first column. From these observations, we propose a compensating circuit to mitigate the voltage variation caused by the wire resistance in a memristor crossbar array.

## 2. Materials and Methods

Figure 1 shows an interesting memristor array circuit for implementing the synaptic weight matrix of a perceptron neural network [21]. A single memristor array and a constant-term circuit are used for realizing the negative and positive synaptic weights, instead of using two complementary crossbar arrays [20,21].

In Figure 1, $g_{j,k}$ is the memristor’s conductance at the crossing point between the *j*th row and the kth column. V_IN,j_ is the input voltage applied to the *j*th row. V_C,k_ is the column-line voltage on the *k*th column. The column line, V_C,F_, is added in Figure 1 instead of using another memristor array [21]. The column line, V_C,F_, is connected to the inputs, from V_IN,1_ to V_IN,m_. In Figure 1, V_C,F_ enters G_F_ that constitutes an inverting OP amp with the negative feedback resistor, R_F1_. The output voltage of G_F_ is V_F_ that is connected to all the column lines from V_C,1_ to V_C,n_ via R_F2_, as shown in Figure 1. By applying Kirchhoff current law to the column line, V_C,F_, we can calculate V_F_ and V_O,k_ with Equations (1) and (2).

(1)$$V_{F}=-\overset{m}{\underset{j=1}{\sum }}\frac{R_{F1}}{R_{B}}V_{\mathrm{IN},j}.$$

(2)$$V_{O,k}=-\left[\overset{m}{\underset{j=1}{\sum }}\left(R_{0}\cdot g_{j,k}\cdot V_{\mathrm{IN},j}\right)+\frac{R_{0}}{R_{F2}}V_{F}\right].$$

If we choose R_F1_ = R_F2_ and combining Equation (1) with Equation (2), the following Equation (3) can be obtained [21].

(3)$$V_{O,k}=-\left[\overset{m}{\underset{j=1}{\sum }}\left(R_{0}\cdot g_{j,k}-\frac{R_{0}}{R_{B}}\right)V_{\mathrm{IN},j}\right].$$

If $-\left(R_{0}\cdot g_{j,k}-\frac{R_{0}}{R_{B}}\right)$ is defined as a synaptic weight of the *j*th row and *k*th column, w*_j_*_,*k*_, we can rewrite Equation (3) with Equation (4).
(4)$$V_{O,k}=\overset{m}{\underset{j=1}{\sum }}w_{j,k}V_{\mathrm{IN},j},$$
where $w_{j,k}=R_{0}\left(\frac{1}{R_{B}}-g_{j,k}\right)=R_{0}\left(\frac{1}{R_{B}}-\frac{1}{M_{j,k}}\right)$.

Equation (4) is used for calculating the output voltage of the *k*th column. The output of each column is a summation of the weighted inputs, hence each column works as a perceptron neuron. In Equation (4), M*_j,k_* is the memristance value of the crossing point between the *j*th row and *k*th column. R_B_ is a constant. The synaptic weight, w*_j,k_*, can be decided to be either negative or positive by adjusting the memristance, M*_j,k_*. The output of the perceptron neuron is decided by a threshold function which produces 0 or 1. By adding the comparator to the output voltage, V_O,*k*_, we can decide if the neuron’s output of the *k*th column, OUT*_k_*, should be activated or not.

(5)$${\mathrm{OUT}}_{k}=\{\begin{array}{l} 1,\;\mathrm{if}\;V_{O,k}\geq V_{\mathrm{REF}}\\ 0,\;\mathrm{if}\;V_{O,k}<V_{\mathrm{REF}} \end{array}.$$

In previous work, the impact of wire resistance is ignored. However, in the crossbar array, the voltage drop along column and row lines cannot be omitted [19,24,25,26,27]. It becomes more serious when the array size increases [24]. The wire resistance between two adjacent junctions is modeled by a small-value resistor, r, as shown in Figure 2.

For the sake of simplicity, in this section we analyze the circuit separately with respect to the wire resistance on horizontal lines and the wire resistance on vertical lines, as shown in Figure 2a,b, respectively. We define V_b1_, V_b2_ as the voltages of node b_1_, b_2_, which are on the first column. Generally, V_b*j*_ is the voltage of node b*_j_* on the first column. Similarly, *V_kj_* is the voltage of node *k_j_*, which is on the *j*th column. Applying Kirchhoff current law for all nodes in Figure 2a, V_F_ and V_O,k_ can be estimated as follows:(6)$$\begin{array}{l} \frac{-V_{F}}{R_{F1}}=\frac{V_{\mathrm{IN},m}-V_{\mathrm{bm}}}{R_{B}}+\ldots +\frac{V_{\mathrm{IN},j}-V_{bj}}{R_{B}}+\ldots +\frac{V_{\mathrm{IN},1}-V_{b1}}{R_{B}}\\ V_{F}=-R_{F1}\left(\overset{m}{\underset{j=1}{\sum }}\frac{V_{\mathrm{IN},j}}{R_{B}}-\overset{m}{\underset{j=1}{\sum }}\frac{V_{bj}}{R_{B}}\right). \end{array}$$
(7)$$\begin{array}{l} \frac{-V_{O,k}}{R_{0}}=\left(V_{\mathrm{IN},m}-V_{km}\right)g_{m,k}+\ldots +\left(V_{\mathrm{IN},j}-V_{kj}\right)g_{j,k}+\ldots +\left(V_{\mathrm{IN},1}-V_{k1}\right)g_{1,k}+\frac{V_{F}}{R_{F2}}\\ V_{O,k}=-R_{0}\left(\overset{m}{\underset{j=1}{\sum }}V_{\mathrm{IN},j}g_{j,k}-\overset{m}{\underset{j=1}{\sum }}V_{kj}g_{j,k}+\frac{V_{F}}{R_{F2}}\right). \end{array}$$

If we assume that R_F1_ = R_F2_, Equation (7) can be simplified as follow:(8)$$V_{O,k}=-\left[\overset{m}{\underset{j=1}{\sum }}\left(R_{0}\cdot g_{j,k}-\frac{R_{0}}{R_{B}}\right)V_{\mathrm{IN},j}-\overset{m}{\underset{j=1}{\sum }}R_{0}V_{kj}g_{j,k}+\overset{m}{\underset{j=1}{\sum }}R_{0}\frac{V_{bj}}{R_{B}}\right].$$

By comparing Equation (8) and Equation (4), we can derive the variation of voltage, ∆V, which is caused by wire resistance on the vertical lines.

(9)$$\Delta V=-\overset{m}{\underset{j=1}{\sum }}R_{0}\frac{V_{kj}}{M_{j,k}}+\overset{m}{\underset{j=1}{\sum }}R_{0}\frac{V_{bj}}{R_{B}}.$$

Here M*_j,k_* is the memristance of the crossing point between the *j*th row and the *k*th column. V_b*j*_ and V*_kj_* are the voltage at nodes b_j_ and k_j_ of the first column and the *k*th column, respectively, as shown in Figure 2a. M*_j,k_* is calculated using Equation (4). It is possible to infer that the variation of voltage presented in Equation (9) can be very small because there are a negative term and a positive term in the right side of Equation (9).

In Figure 2b, wire resistance on vertical lines is omitted whereas wire resistance on horizontal lines is taken into account. The voltages applied to the columns decrease because they are lost from wire resistance. If we define V_j(k)_ as the amount of voltage drop on wire resistance, which is on the *j*th row and between the (*k* − 1)th and *k*th column, the voltage applied to the *j*th row of the *k*th column is calculated as Equation (10).

(10)$$V_{\mathrm{IN},j\left(k\right)}=V_{\mathrm{IN},j}-\overset{k}{\underset{i=1}{\sum }}V_{j\left(i\right)}.$$

Here V_IN,*j*(*k*)_ is the voltage applied to the *j*th row of the *k*th column. The column-line voltage on the *k*th column, V_O,*k*_, can be calculated using Equation (11).

(11)$$V_{O,k}=-\left[\overset{m}{\underset{j=1}{\sum }}\left(R_{0}\cdot g_{j,k}V_{\mathrm{IN},j\left(k\right)}-\frac{R_{0}}{R_{B}}V_{\mathrm{IN},j}\right)\right].$$

By comparing Equation (11) and Equation (3), we obtain the variation of voltage, ∆V*_k_*, of the *k*th column as follows.

(12)$${\Delta V}_{k}=\overset{m}{\underset{j=1}{\sum }}R_{0}g_{j,k}V_{\mathrm{IN},j}-\overset{m}{\underset{j=1}{\sum }}R_{0}g_{j,k}V_{\mathrm{IN},j\left(k\right)}.$$

Calculating V_IN,*j*(*k*)_ by using Equation (10), we obtain ∆V*_k_* as presented in Equation (13).

(13)$${\Delta V}_{k}=\overset{m}{\underset{j=1}{\sum }}\left(R_{0}g_{j,k}\overset{k}{\underset{i=1}{\sum }}V_{j\left(i\right)}\right).$$

Here $\overset{k}{\underset{i=1}{\sum }}V_{j\left(i\right)}$ is the sum of the voltage on k resistors on the *j*th row. Equation (13) indicates that the output voltage of the *k*th column increases because of wire resistance. It is possible to infer that the column close to the first column has small voltage variation and the column far from the first column has large voltage variation. In Equation (13), the voltage variation increases proportionally to the column’s index, *k*. Hence, it is interesting that two adjacent columns can have almost the same amount of voltage variation. Due to this reason, we propose a memristor crossbar array with compensating circuit to mitigate the voltage variation caused by wire resistance. By putting two adjacent columns into a subtraction circuit, the voltage variation can be eliminated significantly. The proposed memristor crossbar is schematically shown in Figure 3.

In Figure 3, the memristor crossbar is composed of 27 columns for recognizing 26 character images. The first column is a constant-term circuit to generate a negative voltage, as mentioned in the previous section. The remaining 26 columns represent 26 perception neurons trained to recognize the 26 characters. The differential amplifies from G_s,2_ to G_s,26_ are inserted into the circuit. The gain of these amplifiers is 1, so they work as the subtractors. The output voltages from V_O,1_ to V_O,n_ are the neuron’s output of columns from Col_1_ to Col_n_. V_O,1_ enter the comparator C_1_ to decide if the neuron’s output of column Col_1_ should be activated or not. V_O,2_ and V_O,1_ go into G_s,2_ that produces V_Os,2_. V_Os,2_ enters the comparator C_2_ to decide if the neuron’s output of column Col_2_ should be activated or not. In general, the output voltage of the column Col*_k_*_−1_ and the column Col*_k_* enter the subtractor G_s,*k*_ for generating the neuron’s output, V_Os,*k*_, of the column Col*_k_*. Using superposition theorem, V_Os,*k*_ can be calculated with the difference of V_O,*k* − 1_ and V_O,*k*_.

(14)$$V_{\mathrm{Os},k}=-V_{O,k-1}\left(\frac{R_{4}}{R_{3}}\right)+V_{O,k}\left(\frac{R_{6}}{R_{5}+R_{6}}\right)\left(\frac{R_{3}+R_{4}}{R_{3}}\right).$$

If we assume that R_3_ = R_4_ = R_5_ = R_6_, we can obtain:(15)$$V_{\mathrm{Os},k}=V_{O,k}-V_{O,k-1}.$$

The differential amplifier is able to reject any signal common to both inputs. That means, if two adjacent columns have almost the same amount of voltage variation, the voltage variation is then mitigated at the output.

The concept of the proposed circuit is shown in Figure 4. The crossbar is trained to recognize the 26 characters from “A” to “Z”. The 25th column is for recognizing the character “Y”. The output of the 25th column is close to 1V when the input is “Y” and close to 0 when the other characters are applied to the input. Similarly, the neuron’s output of the 26th column should be activated if the input is “Z”, as indicated in Figure 4a. In Figure 4b, it is assumed that the wire resistance is present in the crossbar. The output voltage increases as reasoned in the previous section. The two last neurons recognize the input characters incorrectly, as demonstrated in Figure 4b. However, if we put the outputs of two last columns into a subtractor, the voltage variation can be mitigated significantly, as illustrated in Figure 4b. By doing this, we can maintain the recognition rate when wire resistance is present in the crossbar circuit.

## 3. Results

The proposed memristor crossbar circuit in Figure 3 is verified for the application of character recognition. Figure 5a shows eight × eight images of characters used in this simulation. Each character is composed of 64 black-and-white pixels. The proposed memristor crossbar is composed of 64 rows and 27 columns. The first column connects with all inputs through R_B_ to generate the negative voltage as mentioned in the previous section. The remaining 26 columns are for recognition of 26 characters from “A” to “Z”. The 64 input voltages obtained from 64 pixels are applied to the inputs of 64 rows.

The red line in Figure 5b shows a hysteresis behavior of a real memristor based on the film structure of Pt/LaAlO_3_/Nb-doped SrTiO_3_ stacked layer [28]. The black line in Figure 5b represents the behavior model of the memristor used in this paper. This model can well describe various memristive behaviors that come from different kinds of memristors [29]. The circuit simulation is performed using the SPECTRE circuit simulation provided by Cadence Design Systems Inc. Memristors are modeled using Verilog-A and the CMOS technology is given by SAMSUNG 0.13 mm process technology [29,30]. The Verilog-A model parameters are presented in [28]. The wire resistance between two adjacent junctions is set to be 2.5 Ω for a 4F^2^ cross-point structure [19,31]. Figure 6a shows the neuron’s output of the 25th column, which is trained to be activated when character “Y” is applied to the input. Ideally, V_O,25_ is close to 1V for character “Y”, and close to 0V for others. However, the output voltage of the 25th column, V_O,25_, is shifted up because of wire resistance, as reasoned in the previous section. Similarly, in Figure 6b, the neuron’s output of the 26th column is shifted up as a result of the voltage drop along wire resistance. It can be realized that if we compare the column’s output voltage, V_O,26_, with the reference voltage, V_REF_, the neuron’s output of the 26th column can be activated for several input characters, which consequently degrades the recognition rate. The output voltage of the 25th column and the 26th column are put into a subtractor circuit to produce the neuron’s output voltage of the 26th column, V_Os,26_. By doing this, the voltage variation is mitigated significantly, as demonstrated in Figure 6c. When the character “Y” is applied to the inputs, V_Os,26_ is negative, because V_O,25_ is higher than V_O,26_. For the character “Z”, V_Os,26_ is high, as indicated in Figure 6c. The simulation result shown in Figure 6c indicates that the neuron’s output of the 26th column is only activated for the input character “Z”, because the variation of voltage caused by wire resistance is mitigated remarkably by the subtractor circuit.

The proposed circuit is tested with wire resistance that is varied from 0.5 to 2.5 Ω. This range of wire resistance is commonly used and obtained from the International Technology Roadmap for Semiconductors [24,25,31,32,33,34]. Figure 7 shows the comparison of the recognition rate between the memristor crossbar without compensating circuit and the proposed memristor crossbar with compensating circuit when the wire resistance is set to be 0.5, 1.0, 1.5, 2.0, and 2.5 Ω, respectively. The recognition rate of the memristor crossbar without compensating circuit declines dramatically when wire resistance increases. In particular, the recognition rate of the memristor crossbar without compensating circuit is 99%, 95%, and 81%, when the wire resistance is set to be 1.5, 2.0, and 2.5 Ω, respectively. By contrast, the proposed memristor crossbar with compensating circuit can maintain the recognition as high as 100% when wire resistance is as high as 2.5 Ω.

## 4. Discussion

Finally, we discuss the power and area overhead of the proposed memristor crossbar circuit. The proposed circuit uses the compensating circuit constituted by an Op-Amp and four resistors. The proposed circuit consumes more power and area, compared to the memristor crossbar without compensating circuit. However, the proposed memristor crossbar with compensating circuit shows better recognition rate by 19% than the previous memristor crossbar circuit, when wire resistance is set to be 2.5 Ω. Because wire resistance in the crossbar cannot be omitted, the proposed scheme makes the memristor crossbar-based perceptron neural network become more possible. The proposed circuit can be applied to memristor-based crossbar architectures which are used in resistive memory and artificial neural networks [34,35,36].

## 5. Conclusions

In this work, a memristor crossbar-based perceptron neural network with compensating circuit is proposed. The neuron’s outputs of two columns are put into a subtractor circuit to eliminate the voltage variation caused by wire resistance. The memristor crossbar-based perceptron neural network is trained to recognize the 26 characters. The proposed memristor crossbar with compensating circuit shows better recognition rate, compared to the previous memristor crossbar without compensating circuit when wire resistance is taken into account. The simulation result shows that the proposed circuit can maintain the recognition rate as high as 100% when the wire resistance is set to be 2.5 Ω. By contrast, the recognition rate of the memristor crossbar without compensating circuit decreases by 19% when wire resistance is set to be 2.5 Ω.

## Figures and Tables

**Figure 1 micromachines-10-00671-f001:**
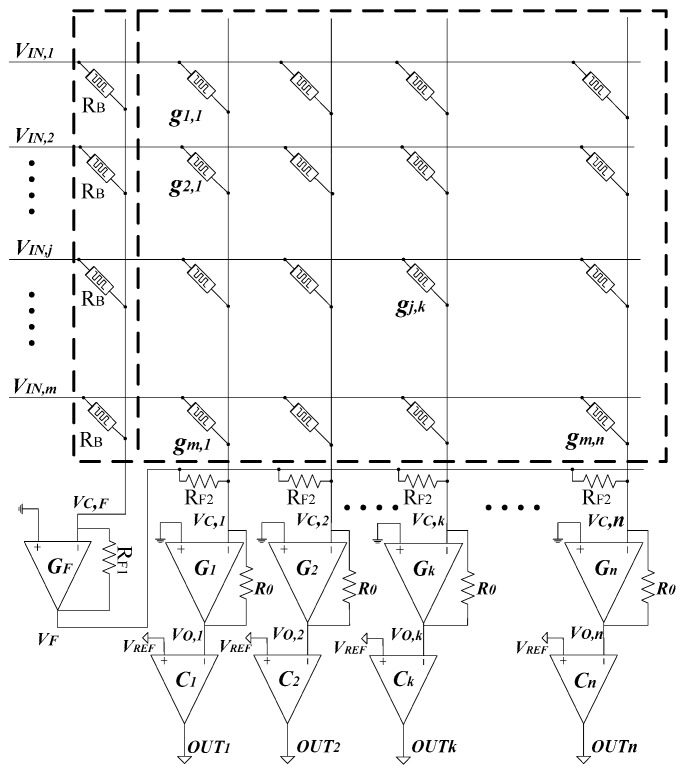
The memristor-based crossbar architecture with a single memristor array and a constant-term circuit for realizing the synaptic matrix of a perceptron neural network [21].

**Figure 2 micromachines-10-00671-f002:**
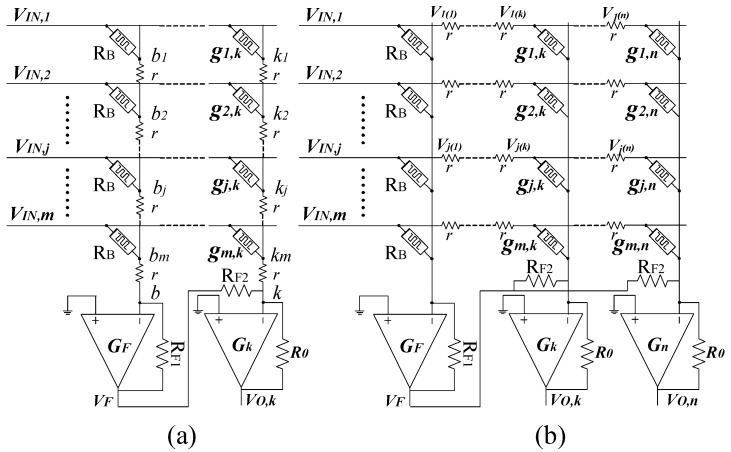
Wire resistance between two adjacent junctions is modeled by a small-value resistor, r, connecting between two crossing points. (**a**) Wire resistance on horizontal lines is omitted. (**b**) Wire resistance on vertical lines is omitted.

**Figure 3 micromachines-10-00671-f003:**
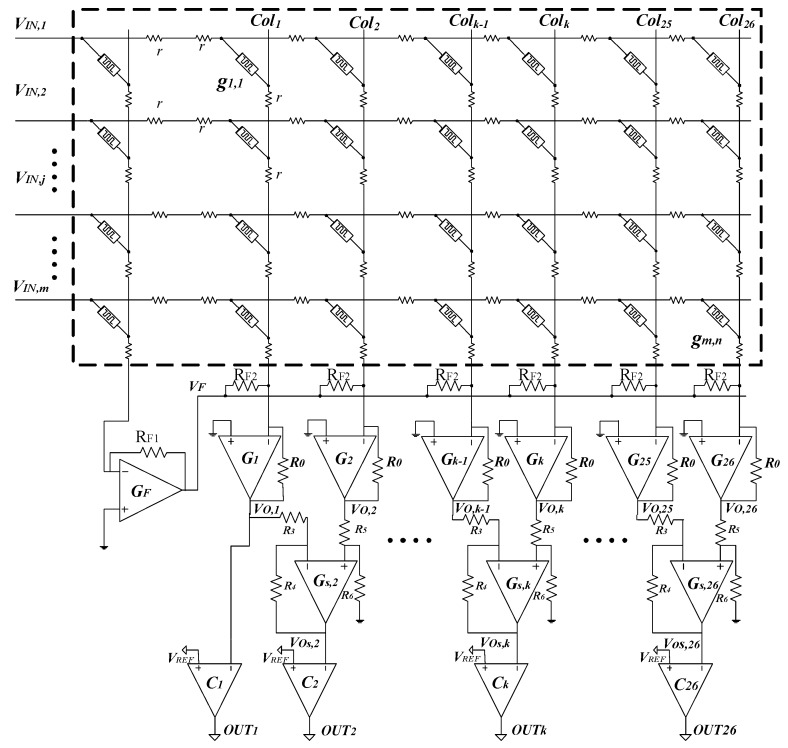
The proposed memristor crossbar with compensating circuit for implementing a perceptron neural network. The outputs of two adjacent columns are put into a differential amplifier working as a subtractor to eliminate the output voltage variation.

**Figure 4 micromachines-10-00671-f004:**
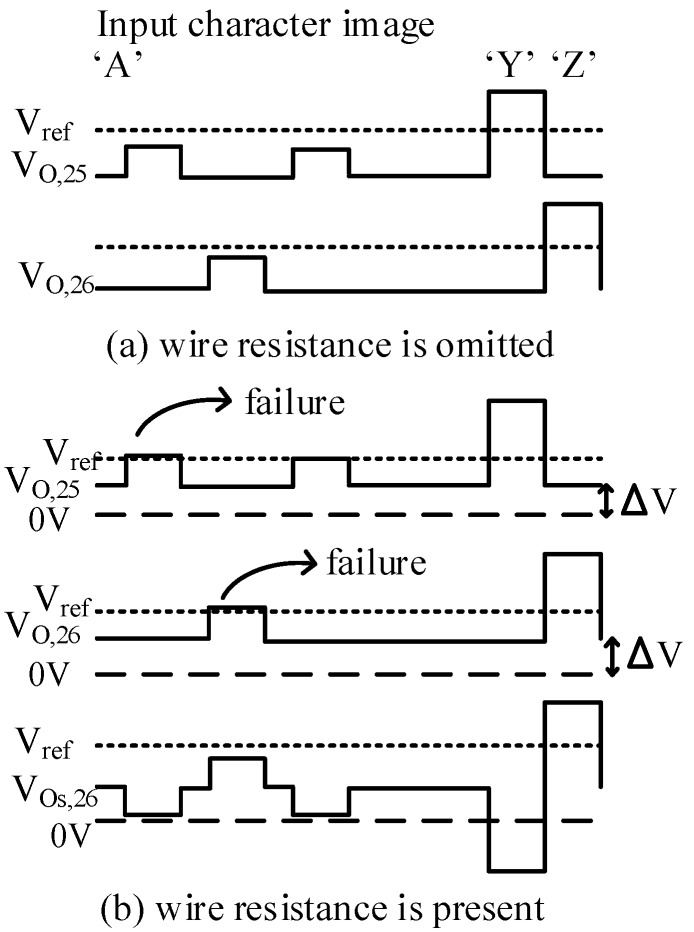
The concept of the proposed circuit for compensating the output voltage variation caused by wire resistance. (**a**) The ideal output of the 25th and 26th columns, which are trained to recognize character images of “Y” and “Z”, respectively. (**b**) The output voltage of the 25th and 26th columns when the wire resistance is taken into account. V_Os,26_ is the output of subtractor for the 26th column, as depicted in Figure 3.

**Figure 5 micromachines-10-00671-f005:**
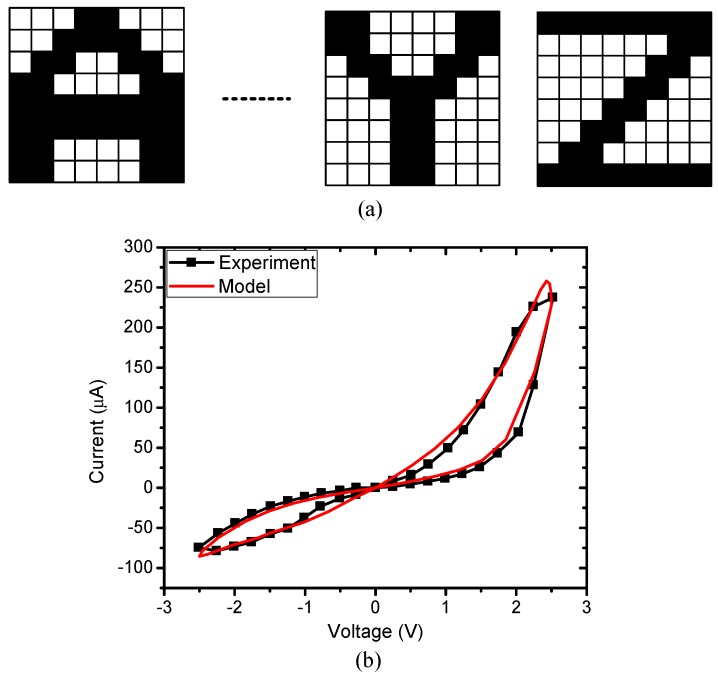
(**a**) The eight × eight pixels images of characters used to test the proposed memristor crossbar circuit. (**b**) The memristor’s current–voltage characteristic measured from the real device and the memristor’s behavior model [28,29].

**Figure 6 micromachines-10-00671-f006:**
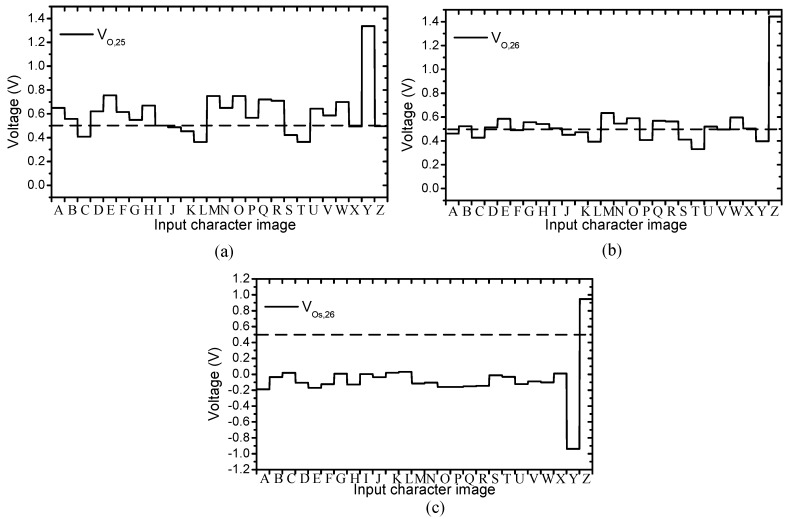
The simulation result of the proposed memristor crossbar array depicted in Figure 4. (**a**) The neuron’s output of the 25th column without compensating circuit. (**b**) The neuron’s output of the 26th column without compensating circuit. (**d**) The neuron’s output of the 26th column with compensating circuit. The wire resistance between two adjacent junctions is set to be 2.5 Ω [19,28].

**Figure 7 micromachines-10-00671-f007:**
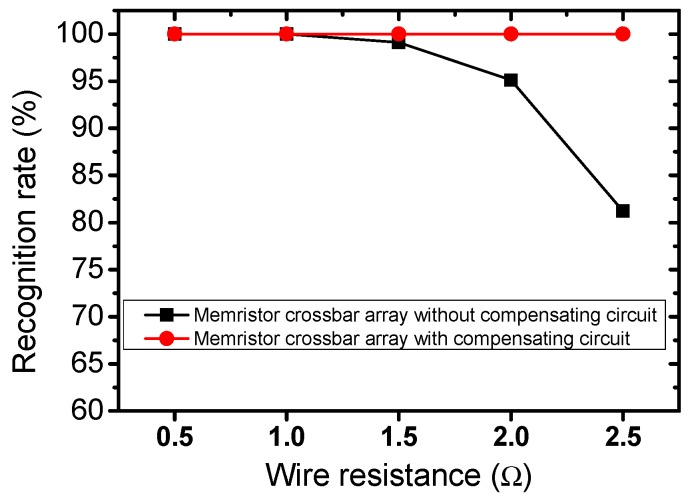
The comparison of the recognition rate between the memristor crossbar without compensating circuit and the proposed memristor crossbar with compensating circuit. The wire resistance is set to be 0.5, 1.0, 1.5, 2.0, and 2.5 Ω, respectively.

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
