# Peer review of "Compensating Circuit to Reduce the Impact of Wire Resistance in a Memristor Crossbar-Based Perceptron Neural Network"

_micromachines, 2019, doi:10.3390/mi10100671_

Round 1

Reviewer 1 Report

I think this is an interesting paper and very well written. The conclusions and the discussion section should probably be merged.

I think the paper can be accepted almost in the present form, otherwise.

On a minor comment

There is a recent review:

https://www.mdpi.com/2227-7080/6/4/118

which discusses also memristors in the analog regime.

I was wondering if the author thinks that the type of circuit he proposes can be used in that case as well (see for instance Section 5.5 and references therein), for instance for learning purposes, as in the paper cited there

https://www.mdpi.com/1099-4300/21/8/789

Author Response

Dear the Reviewer,

Thanks very much for your positive feedback and very nice comment.

As you mentioned a recent review, I think my proposed circuit can be used in memristor based storage and its application in artificial neural network. For this statement, I added one sentence to the section 4 of the revised version. The references were added as well.

The proposed circuit can be applied to memristor-based crossbar architectures which are used in resistive memory and artificial neural network [34-36].

Thank you so much for your work as reviewer.

Reviewer 2 Report

Thanks for the interesting work. 

I have two comments.

1. The authors model the wire resistance as 0.5-2.5 ohm, what do these values mean qualitatively? (What linewidth, process node, how large is each memristor size, array size, etc.)

2. Since this work is based on simulation. It would be better to provide more information on the I-V characteristics of the memristor cell and the Verilog-a model?

Author Response

Dear the Reviewer,

Thanks very much for your positive feedback and very nice comments.

Here is the response of reviewer’s comments

Reviewer’s comments and reply

The authors model the wire resistance as 0.5-2.5 ohm, what do these values mean qualitatively? (What line width, process node, how large is each memristor size, array size, etc.)

 [Reply]

Thank you for your comment.

Wire resistance depends on a variety of factor, such as line width, process node, material, etc. In this work, I do not focus on specific material and technology for crossbar fabrication. Wire resistance can be varied according the factors mentioned above. In the simulation, wire resistance is varied from 0.5Ω to 2.5Ω. This range of wire resistance is very common used in simulation, obtained from the International Technology Roadmap for Semiconductors. To give more information about this, I added following sentences into the section 3 of revised version. The references were added as well. 

The proposed circuit is tested with wire resistance is varied from 0.5Ω to 2.5Ω. This range of wire resistance is common used and obtained from the International Technology Roadmap for Semiconductors [24,25,31-34].     

Since this work is based on simulation. It would be better to provide more information on the I-V characteristics of the memristor cell and the Verilog-a model?

[Reply]

Thank you for very nice comment

I added the Figure 5(b) and the following sentences to provide mode information about I-V characteristic of memristor and the verilog -A model.

The red line in Fig. 5(b) shows a hysteresis behavior of a real memristor based on the film structure of Pt/LaAlO3/Nb-doped SrTiO3 stacked layer [28]. The black line in Fig. 5(b) represents the behavior model of memristor used in this paper. This model can describe well various memristive behaviors that come from different kinds of memristors [29] The circuit simulation is performed using the SPECTRE circuit simulation provided by Cadence Design Systems Inc. Memristors are modeled using Verilog-A and the CMOS technology is given by SAMSUNG 0.13-mm process technology [29,30].The Verilog-A model parameters are presented in [28].

Round 2

Reviewer 2 Report

Overall, the manuscript looks good to me. However, there are still many minor grammar mistakes that need to be addressed.